# A Quantitative Assessment of Methane-Derived Carbon Cycling at the Cold Seeps in the Northwestern South China Sea

**Junxi Feng** [1,2,†], **Niu Li** [3,†], **Min Luo** [4,*], **Jinqiang Liang** [1,*], **Shengxiong Yang** [1], **Hongbin Wang** [1] **and Duofu Chen** [4,5]

[1] MLR Key Laboratory of Marine Mineral Resources, Guangzhou Marine Geological Survey, Guangzhou 510075, China; fengjx123@163.com (J.F.); yangshengxiong@hydz.cn (S.Y.); oceanwang7106@163.com (H.W.)

[2] School of Marine Sciences, Sun Yat-sen University, Guangzhou 510006, China

[3] CAS Key Laboratory of Ocean and Marginal Sea Geology, South China Sea Institute of Oceanology, Chinese Academy of Sciences, Guangzhou 510301, China; liniu@scsio.ac.cn

[4] Shanghai Engineering Research Center of Hadal Science and Technology, College of Marine Sciences, Shanghai Ocean University, Shanghai 201306, China; dfchen@shou.edu.cn

[5] Laboratory for Marine Mineral Resources, Qingdao National Laboratory for Marine Science and Technology, Qingdao 266061, China

[*] Correspondence: mluo@shou.edu.cn (M.L.); ljinqiang@hydz.cn (J.L.); Tel.: +86-021-61900543 (M.L.); +86-020-82253587 (J.L.)

[†] The authors contributed equally to this work.

**Abstract:** Widespread cold seeps along continental margins are significant sources of dissolved carbon to the ocean water. However, little is known about the methane turnovers and possible impact of seepage on the bottom seawater at the cold seeps in the South China Sea (SCS). We present seafloor observation and porewater data of six push cores, one piston core and three boreholes as well as fifteen bottom-water samples collected from four cold seep areas in the northwestern SCS. The depths of the sulfate–methane transition zone (SMTZ) are generally shallow, ranging from ~7 to <0.5 mbsf (meters below seafloor). Reaction-transport modelling results show that methane dynamics were highly variable due to the transport and dissolution of ascending gas. Dissolved methane is predominantly consumed by anaerobic oxidation of methane (AOM) at the SMTZ and trapped by gas hydrate formation below it, with depth-integrated AOM rates ranging from 59.0 and 591 mmol m$^{-2}$ yr$^{-1}$. The $\delta^{13}$C and $\Delta^{14}$C values of bottom-water dissolved inorganic carbon (DIC) suggest discharge of $^{13}$C- and $^{14}$C-depleted fossil carbon to the bottom water at the cold seep areas. Based on a two-endmember estimate, cold seeps fluids likely contribute 16–26% of the bottom seawater DIC and may have an impact on the long-term deep-sea carbon cycle. Our results reveal the methane-related carbon inventories are highly heterogeneous in the cold seep systems, which are probably dependent on the distances of the sampling sites to the seepage center. To our knowledge, this is the first quantitative study on the contribution of cold seep fluids to the bottom-water carbon reservoir of the SCS, and might help to understand the dynamics and the environmental impact of hydrocarbon seep in the SCS.

**Keywords:** porewater geochemistry; bottom seawater; methane-derived carbon cycling; cold seeps; South China Sea

## 1. Introduction

The continental margin sediments contain large reservoirs of methane either as dissolved phase in porewater, solid gas hydrate or free gas (bubbles) depending on its in situ solubility. Submarine cold

seeps refer to the leakage of methane-rich fluids out of the sedimentary column, which is widespread along continental margins and serve as a significant component of the global carbon cycle [1,2]. Nowadays hundreds of marine cold seeps are known worldwide, which can be windows to various submarine geospheres with different depth levels. Subduction zones and organic-rich passive margins host most of the cold seeps globally. The source of seep fluids range from tens of meters (groundwater aquifers) to tens of kilometers (subducted oceanic plates) below the seabed [2]. The most important biogeochemical process is anaerobic oxidation of methane (AOM: $CH_4 + SO_4^{2-} \rightarrow HCO_3^- + HS^- + H_2O$) through a syntrophic consortium of methanotrophic archaea and sulfate-reducing bacteria. Before the upward dissolved methane being transported into the water column, it is mostly consumed by AOM at the sulfate–methane transition zone (SMTZ) [3,4]. Through this reaction methane is converted to dissolved inorganic carbon (DIC) which could be partially removed from solution by authigenic carbonate precipitation [5]. Therefore, AOM is one of the most important biogeochemical processes linking the biogeochemical sulfur and carbon cycles in marine sediments, and serves as a barrier to prevent methane from subsurface from entering into the hydrosphere [6,7].

Excluding seep systems, global estimates show that AOM within the SMTZ on the large continental slopes consumes around 0.05 Gt methane-derived carbon annually [6]. In comparison, methane flux above the gas hydrate occurrence zone (GHOZ) is estimated to be up to 0.03 Gt yr$^{-1}$ at the scattered hotspot-like cold seeps, whereas seeps could emit 0.02 Gt C as methane to the hydrosphere annually [6]. The emitted $CH_4$ and $H_2S$-rich fluids could support unique chemosynthetic ecosystems, ranging from microbes to megafauna, at or near the sediment–water interface (SWI) [4,8,9]. Additionally, the submarine cold seeps could also contribute considerable amounts of fossil carbon in the forms of DIC and dissolved organic carbon (DOC) to the water column [10–14]. It is thus hypothesized that carbon released by seeps may aggravate ocean acidification and de-oxygenation and even contribute to abrupt climate change (e.g., the Paleocene-Eocene thermal maximum) [15–17]. Although the discoveries of cold seeps in the global continental margin have been made for over thirty years [18], the quantitative regional and global estimation of methane fluxes and depth-integrated methane turnovers from cold seeps remains fragmentary [6,19,20]. The reasons for this include the uncertainties of seep distribution, temporal and spatial variability in seep intensity and activity, together with the physical and biogeochemical processes modulating methane seepage [20].

Methane seepages are widespread on the northern slope of South China Sea (SCS) as revealed by authigenic carbonates, seep-associated fauna as well as anomalous porewater and sediment geochemistry influenced by fluid seepage at more than thirty sites [21]. Seafloor features associated with seeps in the SCS include mud volcanoes, pockmarks and carbonate deposits [21]. Recently, Zhang et al. (2019) used a dataset of the biogeochemical rates and fluxes of methane and DIC at 395 sites at cold seeps and hydrate-bearing areas in the northern SCS to extrapolate the areal rates and fluxes by spatial interpolation [22]. The results revealed that the rates of AOM and carbonate precipitation and effluxes of methane and DIC in Qiongdongnan area were at least one order of magnitude higher than those in Dongsha and Shenhu, due to the occurrence of intensive methane gas bubbling [22]. Moreover, most DIC generated by AOM is diffused into the bottom seawater in the northern SCS. Compared to other cold seeps worldwide, the biogeochemical rates in the northern SCS are generally lower than those in active continental margins and euxinic environment (e.g., the Black Sea), but are similar to those in passive continental margins [22]. Owing to the high spatial heterogeneity in the methane turnovers at seeps, it is necessary to carry out more investigations on the geochemistry of dissolved species in different habitats within the seeps in the Qiongdongnan area to better constrain the methane budget.

The active Haima cold seeps were recently discovered on the northwestern slope of Qiongdongnan Basin in the SCS [23]. Several sites with gas bubbling or shallow gas hydrates were also identified by hydroacoustic imaging and/or sediment coring around this area [24–28]. Recent studies have shown pronounced spatial and temporal changes in the intensity of methane seepage and the discharge of considerable amount of methane and DIC to the water column around the cold seeps [29–31]. In this study, we present seafloor observations and porewater geochemical data of six push cores, one piston

core and three boreholes as well as fifteen bottom-water samples collected at two newly discovered seeps with weak activities and an intensely active seep in the Qiongdongnan Basin. Using steady-state reaction-transport modeling, we aim at quantifying the methane turnover rates in shallow sediments at the seep sites. The $\delta^{13}C_{DIC}$ and $\Delta^{14}C_{DIC}$ values of bottom-water samples are also measured to investigate the potential contribution of fossil carbon released by cold seeps. Furthermore, a two-endmember simple mass balance model is applied to evaluate the contribution of cold seep fluids in this area. In combination with seafloor observations and geochemistry of fluids from cold seeps, we demonstrate the heterogeneity of fluxes and geochemical processes at the cold seep areas of the Qiongdongnan Basin.

## 2. Geological Background

The northern SCS is characterized as a Cenozoic passive continental margin [32]. The Qiongdongnan Basin, located in the northwestern part of the SCS, is a northeastern trended Cenozoic sedimentary basin, which is covered by thick sedimentary materials [33]. This basin has experienced a rifting stage and a post-rift thermal subsidence stage. During the first stage, plenty of half-grabens and sags were formed in the basin. At the later stage, thermal subsidence occurred and thick sediment sequences mainly consisting of mudstones were deposited since Miocene. Consequently, the sedimentation rates and the geothermal gradients are both high in the Qiongdongnan Basin [34]. As a result, there are abundant hydrocarbon accumulations in this basin. Widespread faulting and/or diapirism facilitated the migration of hydrocarbon gas to the upper strata [35]. Gas hydrate reservoirs were identified via numerous bottom-simulating reflectors (BSRs) and gas chimneys as the main types of fluid conduit in this region [36–38].

The Haima cold seeps have been discovered in the southern uplift belt of the Qiongdongnan Basin with water depths of ~1400 m during R/V Haiyang-6 cruises in 2015 and 2016. Abundant chemosynthetic communities, methane-derived authigenic carbonates and massive gas hydrates in the near-surface sediments were observed within the seepage area [23]. Seafloor observations and sampling were conducted at sites HM-ROV05 and HM-ROV within the Haima cold seeps. In addition, another two weak seeps (QH-ROV05 and QH-ROV07) were also dived during R/V Haiyang-6 cruises in 2018 (Figure 1). These seeps are located above two acoustic blanking zones that were inferred as gas chimneys to the northeast of the Haima cold seeps (Figure 1). The bathymetry in these areas is characterized by flat topography with water depths ranging from 1700 to 1800 m (Figure 1). During a later gas hydrate drilling expedition, multiple gas hydrates were observed from 8 to 174 mbsf of site QH-W08-2018 (hereinafter referred to as site W08, including two nearby holes W08B and W08C) within the investigation area QH-ROV07 and from 15 to 160 mbsf of site QH-W09-2018 (hereinafter referred to as W09) within the area QH-ROV05, respectively [28,39,40].

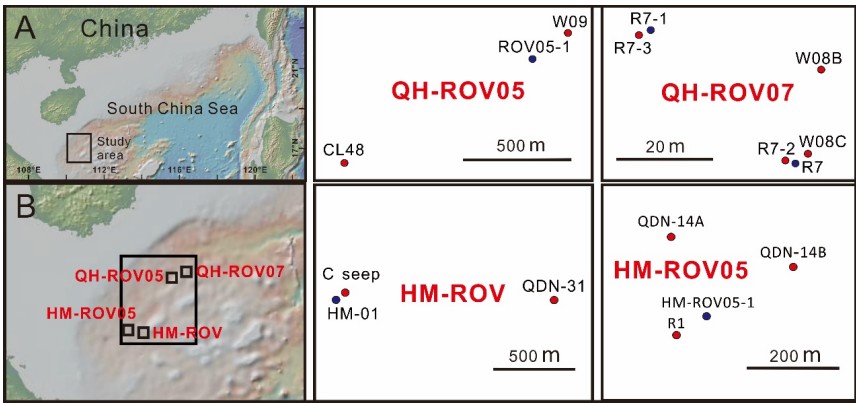

**Figure 1.** Location of the study area (black rectangle and square) in South China Sea (**A**,**B**) and locations of the cores and bottom-water samples collected within cold seep areas (red and blue dots) The samples collected within the remotely-operated vehicle (ROV) investigation stations (blue dots) include cores and bottom-water samples. The site locations of R1, QDN-14A, QDN-14B and QDN-31 are adjusted from [31].

## 3. Materials and Methods

### 3.1. Seafloor Observations

Two remotely-operated vehicle (ROV) dives were conducted around sites W08 and W09 by ROV "Haima" in April 2018 and Fugro ROV "FCV 2000D" in June 2018. Another dive was conducted in May 2018 at the seep site C using the ROV "Haima". Photos of authigenic carbonates and chemosynthetic communities and/or gas bubbles at these sites were taken during these cruises. Moreover, methane concentrations in bottom seawater were measured using a methane sensor mounded on the ROV "Haima".

### 3.2. Sampling and Analytical Methods

During the R/V Haiyang-6 cruise, 6 push cores (~70 cm) and 15 bottom-water samples were collected from sites QH-ROV05, QH-ROV07, HM-ROV05 and HM-ROV using the ROV "Haima" in 2018 (Figure 1, Table 1). In QH-ROV07, 2 push cores, R7 and R7-2, were recovered at the edge of an authigenic carbonate deposit, where the gas hydrate-bearing core W08C was drilled only ~5 m away. Another hydrate-bearing core W08B was drilled ~20 m away from the carbonate deposit where a dome-like structure and fragments of recently dead seep-associated bivalves were observed. Two other push cores, R7-1 and R7-3, were taken ~40 m away from W08B. In QH-ROV05, a push core ROV05-1 and a piston core (CL48) were also collected about 200 m and 1.1 km away from site W09, respectively (Figure 1). In HM-ROV, a push core HM-01 was collected near the seep C with acoustic flare (Figure 1). No sediment core was collected at site HM-ROV05 due to a lack of time during that cruise.

**Table 1.** Information on the studied push cores and piston cores from the Qiongdongnan Basin.

| Site | Water Depth (m) | Seafloor Temperature (°C) | Core Length (cm) |
|---|---|---|---|
| QH-ROV05 | 1722 | 2.2 | R5-1(78 cm), R5-2 (50 cm), QH-CL48 (763 cm) |
| QH-ROV07 | 1737 | 2.2 | R7 (60 cm), R7-1 (60 cm), R7-2 (50 cm), R7-3 (78 cm) |
| HM-ROV | 1405 | 2.9 | HM-1(70 cm) |

On the other hand, bottom-water samples, including R01-2018, ROV07, R-07, R-07-1 around W08, R-05-shell and ROV05 around W09, HM-R003-1 from HM-ROV05, HM-2-vent and HM-3-vent from HM-ROV were collected using specially-made fluid samplers which are temperature-held and pressure-tight ~5 m above the seafloor. Other bottom-water samples, including ROV05-1 from W09, HM-1 from HM-ROV05 as well as HM-2, HM-3 from HM-ROV, were collected by Niskin bottles ~5 m above the seafloor. In addition, three other samples, including ROV07+v, ROV07-1 from W08 and ROV05-1 from W09, were collected from the top of push cores (Table 1).

The recovered cores were immediately brought to the onboard laboratory for porewater extraction. The porewater samples of the push cores and the piston core were collected at 5 or 10 cm intervals and at 60 cm intervals using Rhizon samplers, respectively. Porewater samples were acidified with ultra-pure concentrated $HNO_3$ (10 μL $HNO_3$ per 1 mL sample) for determining the concentrations of major elements onshore. Porewater samples for analysis of the concentration and carbon isotopic composition of DIC were preserved with a saturated $HgCl_2$ solution (~20 μL $HgCl_2$ per 5 mL of sample). All the porewater samples were stored at 4 °C until further analysis. In addition, the porewater $PO_4^{3-}$ concentrations of the piston core CL48 were measured onboard using the spectrophotometric method according to [41] with a UV–Vis Spectrophotometer (Hitachi U5100, Hitachi Limited, Tokyo, Japan). The precision for phosphate was ±3.0%. The total alkalinity (TA) of CL48 was determined onboard by direct titration with ~0.006 M HCl using a pH meter. The analysis was calibrated using the seawater standard of International Association for the Physical Sciences of the Oceans (IAPSO), with a precision and detection limit of 0.05 meq $L^{-1}$. For the bottom-water samples, total alkalinity (TA) contents were determined onboard by the same method.

Moreover, porewater sampling and analysis of sulfate ($SO_4^{2-}$) concentrations in the porewater samples of W08 and W09 were reported in [28]. Calcium ($Ca^{2+}$) concentrations were also measured aboard via ion chromatography (Dionex ICS-2100, Thermo Fisher Scientific, Waltham, MA, USA). The 4 ml sediment samples of W08 and W09 were sealed in 26 mL glass vials onboard. The vials were placed in a 60 °C oven for 30 min to fully release the hydrocarbon gases weakly adsorbed on the surfaces and pores of the sediment particles. After that, the concentrations of hydrocarbon gas in the headspace vials were measured onboard using the gas chromatograph method (Inficon Fusion MicroGC). The detection limit for all gases was 5 ppm.

For the porewater samples from the push and piston cores, $SO_4^{2-}$ and $Ca^{2+}$ concentrations were measured on a Dionex ICS-5000+ ion chromatograph (analytical precision of <2%) at the South China Sea Institute of Oceanology, Chinese Academy of Sciences. The anion ($SO_4^{2-}$) and cation ($Ca^{2+}$) concentrations were determined by 500- and 100-fold dilution, respectively, using ultra-pure water. Concentration and isotopic analyses of DIC were determined using a Thermo Finnigan Gas Bench coupled with a Thermo Finnigan Delta V Advantage at the Louisiana State University. The analytical precisions were better than 0.1‰ for $\delta^{13}C$. For the porewater samples from W08 and W09, DIC concentrations and $\delta^{13}C_{DIC}$ values were determined via a continuous flow mass spectrometer (Thermo Delta V Advantage). The analytical precisions were better than 0.2‰ for $\delta^{13}C$. All $\delta^{13}C$ data in this study are reported in per mil (‰) using the δ notation, relative to the standards Vienna-Pee Dee Belemnite (V-PDB).

The natural radiocarbon $^{14}C$ contents of DIC ($\Delta^{14}C_{DIC}$) for bottom-water samples were obtained at Beta Analytic Inc., Miami, FL, USA, using accelerator mass spectrometry (AMS). Sample pretreatment, preparation, and measurement were conducted at Beta Analytic Inc. The Cambridge half-life (5730 ± 40 years) was used to calculate the apparent radiocarbon age and $\Delta^{14}C$ [42,43]. The accuracy and precision of the $\Delta^{14}C_{DIC}$ measurements was 0.1‰ and 1.7‰, respectively.

*3.3. Reaction-Transport Model*

In this study, a one-dimensional, steady-state reaction-transport model was applied to simulate one solid (POC) and four dissolved species including sulfate ($SO_4^{2-}$), methane ($CH_4$), dissolved inorganic carbon (DIC) and calcium ($Ca^{2+}$). The model is modified from previous simulations of methane-rich sediments [20,44–46], and a full description of the model is shown in the Appendix A. All the reactions considered in the model and the expressions of kinetic rate are listed in Table 2.

**Table 2.** Rate expressions of the reactions considered in the model.

| Rate | Kinetic Rate Law* |
|---|---|
| Total POC degradation (wt.% C yr$^{-1}$) | $R_{POC} = \left(0.16 \cdot \left(a_0 + \frac{x}{v_s}\right)^{-0.95}\right) \cdot POC$ |
| POM degradation via sulfate reduction (mmol cm$^{-3}$ yr$^{-1}$ of $SO_4^{2-}$) | $R_{SR} = 0.5 \cdot R_{POC} \cdot \frac{[SO_4^{2-}]}{[SO_4^{2-}] + K_{SO_4^{2-}}} / f_{POC}$ |
| Methanogenesis (mmol cm$^{-3}$ yr$^{-1}$ of $CH_4$) | $R_{MG} = 0.5 \cdot R_{POC} \cdot \frac{K_{SO_4^{2-}}}{[SO_4^{2-}] + K_{SO_4^{2-}}} / f_{POC}$ |
| Anaerobic oxidation of methane (mmol cm$^{-3}$ yr$^{-1}$ of $CH_4$) | $R_{AOM} = k_{AOM} \cdot [SO_4^{2-}][CH_4]$ |
| Authigenic carbonate precipitation (mmol cm$^{-3}$ yr$^{-1}$ of $Ca^{2+}$) | $R_{CP} = k_{Ca} \cdot \left(\frac{[Ca^{2+}] \cdot [CO_3^{2-}]}{K_{SP}} - 1\right)$ |

*Notation: $R_{POC}$ (wt.% C yr$^{-1}$) is the POC degradation rate, $a_0$ (yr) is the initial age of organic matter in surface sediments, $v_s$ (cm yr$^{-1}$) is the burial velocity of solids, $x$ (cm) is the depth in the sediment, $Kc$ is an inhibition constant for POC degradation due to DIC and $CH_4$ accumulation in the porewater, POC (wt.%) is the POC content in sediments. $R_{SR}$ (mmol cm$^{-3}$ yr$^{-1}$ of $SO_4^{2-}$) is the rate of sulfate reduction, $[SO_4^{2-}]$ is the $SO_4^{2-}$ concentration, $K_{SO_4^2}$ is the Michaelis–Menten constant for the inhibition of sulfate reduction at low sulfate concentrations, $f_{POC}$ converts between POC (dry wt.%) and DIC (mmol cm$^{-3}$ of porewater): $f_{POC} = MW_C/10\Phi/(1 - \Phi)/\rho_S$, where MWC is the molecular weight of carbon (12 g mol$^{-1}$), $\rho_S$ is the density of dry sediments, and $\Phi$ is the porosity. $R_{MG}$ (mmol cm$^{-3}$ yr$^{-1}$ of $CH_4$) is the rate of methanogenesis. $R_{AOM}$ (mmol cm$^{-3}$ yr$^{-1}$ of $CH_4$) is the rate of AOM, $k_{AOM}$ is the rate constant, $[CH_4]$ is the concentration of dissolved $CH_4$. $R_{CP}$ (mmol cm$^{-3}$ yr$^{-1}$ of $Ca^{2+}$) is the rate of authigenic carbonate precipitation, $k_{Ca}$ (mol·cm$^{-3}$·yr$^{-1}$) is the rate constant, $K_{SP}$ (mol$^2$·L$^2$) is the thermodynamic equilibrium constant, $[Ca^{2+}]$ and $[CO_3^{2-}]$ are the concentrations of $Ca^{2+}$ and $CO_3^{2-}$, respectively.

Due to the porewater sampling resolution, any influence of bioturbation and bioirrigation on the porewater profiles cannot be resolved. Therefore, solutes are assumed to be mainly transported by molecular diffusion and porewater burial, while solid species are assumed to be transported only by burial with prescribed compaction. At sites W08C and W09, the porewater species concentrations were close to that of seawater in the upper ~2 and ~3 m, respectively. This feature may be attributed to the bubble irrigation which is often observed at cold seep sites [20,30,45]. The porewater mixing with bottom water induced by rising gas bubbles can be described as a nonlocal transport similar to bioirrigation.

$$R_{Bui} = \alpha_0 \cdot \frac{\exp\left(L_{irr} - \frac{x}{\alpha_1}\right)}{1 + \exp\left(L_{irr} - \frac{x}{\alpha_1}\right)} \cdot (C_0 - C_x), \tag{1}$$

where $\alpha_0$ (yr$^{-1}$) is the coefficient of irrigation intensity, $L_{irr}$ (cm) is the depth of bubble irrigation, $\alpha_1$ (cm) is the parameter determining how quickly bubble irrigation is attenuated to zero at an approximate depth of $L_{irr}$, $C_0$ is the solute concentration at the SWI, and $C_x$ is the concentration at any depth within the irrigation zone.

Major biogeochemical reactions considered in the model are particulate organic matter (POM) degradation via sulfate reduction, AOM, methanogenesis and authigenic carbonate precipitation. Organic matter mineralization via aerobic respiration, denitrification, and metal oxide reduction were ignored because these processes mainly occur in the surface centimeter-scale sediments, which cannot be resolved by our sampling resolution.

The total rate of POM mineralization, $R_{POC}$ (wt.% C yr$^{-1}$), is calculated via the power law model in which the initial age of organic matter in surface sediments, $a_0$ (yr) is considered in surface sediments [47]. $a_0$ is constrained using the measured PO$_4$$^{3-}$ concentrations of the reference core.

When sulfate is almost completely consumed, the remaining POM is degraded to $CO_2$ and $CH_4$ via methanogenesis:

$$2CH_2O(POP)_{rP} \rightarrow CO_2 + CH_4 + 2r_P PO_4{}^{3-}. \tag{2}$$

The main pathways of methanogenesis in marine sediments are organic matter fermentation and $CO_2$ reduction [48]. Their net reactions at steady state are balanced as equivalent amounts of $CO_2$ and $CH_4$ are produced when per mole of POM is degraded [49]. Accordingly, the reaction of methanogenesis is a net reaction.

Methane is considered to be consumed by AOM [3]:

$$CH_4 + SO_4{}^{2-} \rightarrow HCO_3{}^- + HS^- + H_2O. \tag{3}$$

The rate constant for AOM, $k_{AOM}$ (cm$^3$ mmol$^{-1}$ yr$^{-1}$), is tuned to the sulfate profiles within the SMTZ.

The loss of Ca$^{2+}$ resulting from the precipitation of authigenic carbonates as calcite (Ca$^{2+}$ + HCO$_3$$^-$ → CaCO$_3$ + H$^+$) was simulated in the model using the thermodynamic solubility constant as defined by [50] (Table 2). A porewater pH value of 7.3 was used to calculate CO$_3$$^{2-}$ from modeled DIC concentrations [51]. CaCO$_3$ was not simulated explicitly in the model.

The length of the simulated model domain was set to 1000 cm for W08B and W08C, 2000 cm for W09, 500 cm for R7-1 and R7-3 and HM-1. Upper boundary conditions for all species were imposed as fixed concentrations (Dirichlet boundary) using measured values in the uppermost sediment layer where available. A zero concentration gradient was imposed at the lower boundary for all the species except CH$_4$. CH$_4$ concentration at the lower boundary was a tunable parameter constrained from the SO$_4$$^{2-}$ profile. The model was solved using the NDSolve object of MATHEMATICA V. 10.0 (Wolfram Research, Champaign, IL, USA). The steady-state simulations were run for 10$^7$ years to achieve the steady state with a mass conservation of >99 %. Further details on the model solutions can be found in Appendix A.

## 4. Results

### 4.1. Site Characteristics

Seafloor observations showed that a small mud mound sparsely colonized by living clams at site QH-ROV05 (Figure 2). At site QH-ROV07, massive authigenic carbonate deposits and dead clams were observed on the seabed (Figure 2B,C). Gas hydrate-bearing core W08B was drilled ~20 m away from the carbonate deposit where there is a dome-like structure (Figure 2D). Fragments of recently dead seep-associated bivalves and tubeworms (Figure 2D) were scattered on the periphery of the dome. Scatter seep-associated bivalve fragments were observed on the seafloor. The sediments of the cores R7-1 and R7-3 mainly consist of black-green silty clay with some shell fragments and a strong odor of hydrogen sulfide.

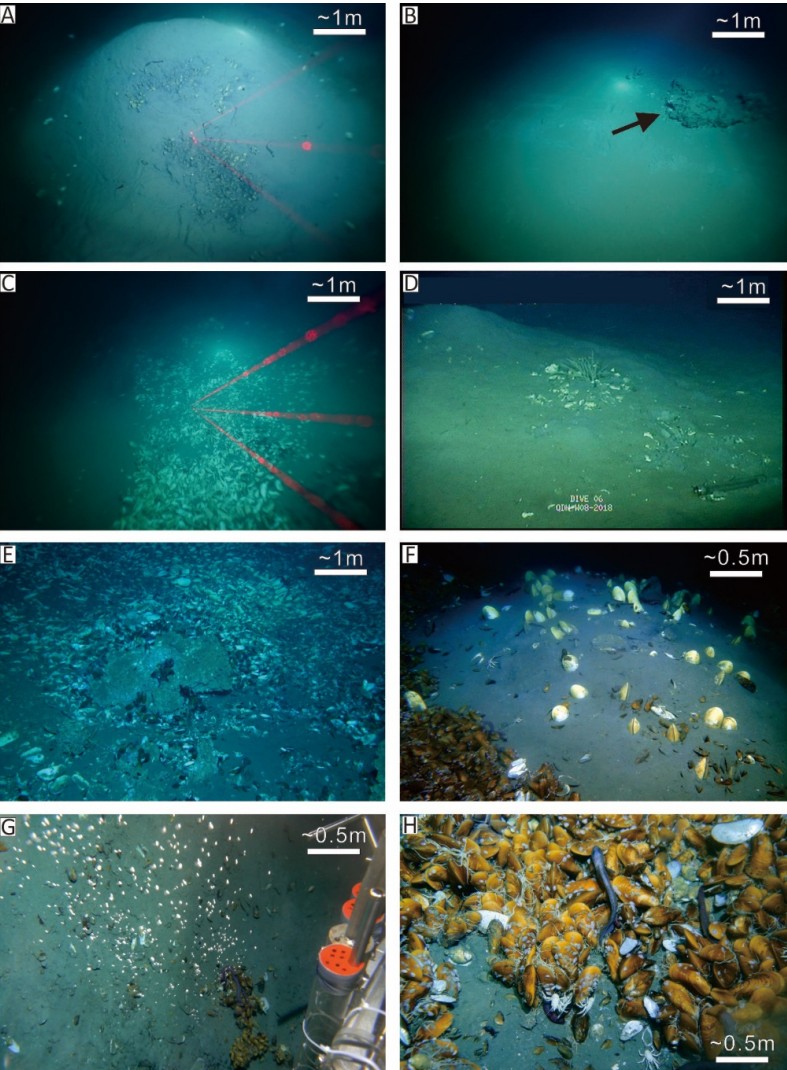

**Figure 2.** Seafloor observations of cold seep areas QH-ROV05 (**A**), QH-ROV07 (**B–D**), HM-ROV05 (**E**) and HM-ROV (**F–H**). (**A**) A small mud mound where living clams were sparsely colonized. The distance between two laser beams was 10 cm. (**B**) A massive authigenic carbonate deposit (black arrow) observed by ROV "Haima". (**C**) Bivalve fragments on the seabed (adapted from [40]). (**D**) Bivalve fragments and putative tubeworms (adapted from [40]). (**E**) Extensive authigenic carbonate pavements with bacterial mats, squat lobsters and a small amount of living tubeworms. (**F**) Dead bivalves scattered on a mud mound. (**G**) Gas bubbling on a mud mound. (**H**) Abundant deep-sea mussels clustered on the flank of a mud mound.

At site HM-ROV05, there were extensive authigenic carbonate pavements with bacterial mats, squat lobsters and a small amount of living tubeworms colonizing the fractures (Figure 2E). Dead bivalves were scattered and gas bubbling was observed on the mud mounds (Figure 2F,G). Abundant deep-sea mussels were clustered on the flank of the mud mounds (Figure 2H). Methane concentrations in bottom water were <5.4 nM at site QH-ROV07, 5.4–36 nM at site QH-ROV05, 5.4–900 nM at site HM-ROV05 and >5.4 × 10$^4$ nM at site HM-ROV, respectively.

## 4.2. General Geochemical Trends

Profiles of porewater $SO_4^{2-}$, DIC, $Ca^{2+}$ concentrations are shown in Figures 3–5, Tables A3 and A4. In the borehole W08B, the $SO_4^{2-}$ concentrations were nearly depleted near the seafloor, whereas DIC concentrations were high and $Ca^{2+}$ concentrations were depleted from the seabed below. In the borehole W08C, the $SO_4^{2-}$ concentrations were depleted below 8 mbsf, which indicated the depth of SMTZ was shallow and located at ~2–8 mbsf. Due to the coarse sampling resolution in the upper 10 m of the hydrate-bearing borehole W08C, no clear downcore trend of porewater geochemical data was observed (Table A3). Considering the close proximity between the push cores (R7 and R7-2) and W08C, we present the porewater data of these cores in one panel for the sake of modelling. In contrast, both R7 and R7-2 do not show significant downcore variations in $SO_4^{2-}$, DIC and $Ca^{2+}$ concentrations. $SO_4^{2-}$ concentrations decreased quasi-linearly from 23.1 to 9.5 mM at HM-1 and from 23.5 to 18.0 mM at R7-1. At the core R7-3, $SO_4^{2-}$ concentrations decreased from 24.5 to 21.7 mM at 0.4 mbsf and an abrupt reversal in concentration gradient of $SO_4^{2-}$ occurred below. $Ca^{2+}$ concentrations showed similar trends as $SO_4^{2-}$, with sharper gradient at HM-1 and R7-1 than at R7-3 (Figures 3–5, Table A4). DIC concentrations showed downcore increase trend that were opposite to $SO_4^{2-}$ profiles (Figures 3–5, Table A4).

Moreover, at the borehole W09, the $SO_4^{2-}$, DIC and $Ca^{2+}$ concentrations were also close to seawater values in the upper 3.5 m and began to change below. The $SO_4^{2-}$ concentrations were depleted below 8.5 mbsf, which indicated the depth of SMTZ was shallow and located at ~5–8.5 mbsf. In addition, dissolved $SO_4^{2-}$, $Ca^{2+}$ and DIC concentrations exhibit near-seawater values with depth at the cores R05-1and CL48 (Figures 3–5, Table A3). The profiles $\delta^{13}C_{DIC}$ values mirrored those of DIC concentrations reaching minimum at the SMTZ of the cores and shifting to positive values in the methanogenic zone as shown in the porewater profiles of W08B, W08C and W09 (Figures 3–5, Table A3). The $\delta^{13}C_{DIC}$ values of R7 and R7-2 showed slightly downcore decrease, whereas those of R05-1 and R05-2 display near-seawater values with depth (Figures 3–5, Table A4).

Natural radiocarbon measurements of DIC for bottom-water samples yielded the $^{14}C$ ages of DIC ranging from 1250 to 590 years BP ($\Delta^{14}C$ = −151‰ to −71‰). The $\Delta^{14}C_{DIC}$ values of temperature- and pressure-tight (T,P-tight) samples range from −149‰ to −95‰ (Table A5). Moreover, the $\delta^{13}C_{DIC}$ values of T,P-tight samples (−3.4‰ to −1.6‰) were generally lower than those of T,P-tight free samples (−2.1‰ to −1.7‰). Small variations were measured in pH values and total alkalinity (TA) concentrations of the bottom-water samples (values ranged from 7.6 to 7.9 for pH and ranged from 2.8 to 3.2 mM for TA) (Table A5).

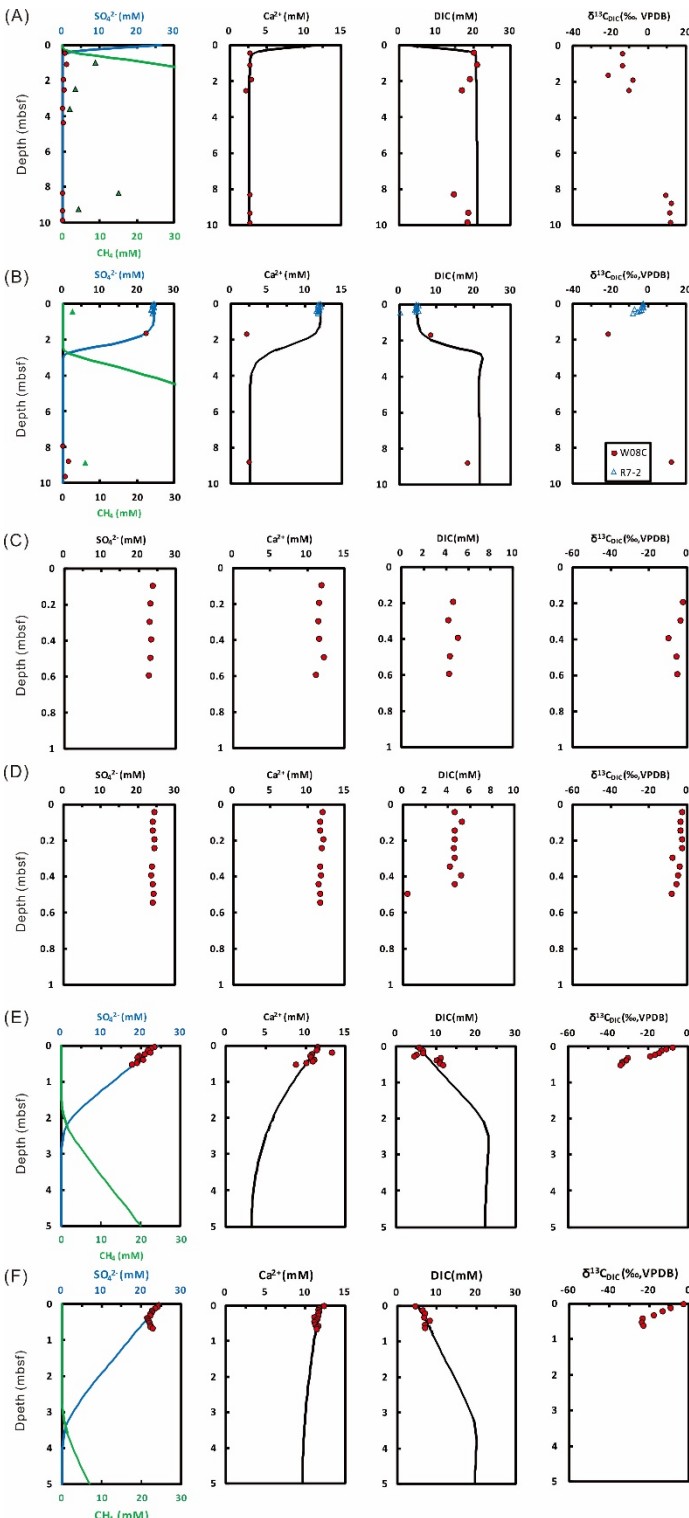

**Figure 3.** Measured (dots) and simulated (curves) depth profiles of cores in the seep area W08. Down-depth concentration of $SO_4^{2-}$, $Ca^{2+}$, DIC, $\delta^{13}C_{DIC}$ and/or $CH_4$ of (**A**) W08B and (**B**) W08C above 10 mbsf, (**C**) R7, (**D**) R7-2, (**E**) R7-1 and (**F**) R7-3 are shown.

### 4.3. Reaction-Transport Modeling

The simulation profiles and methane turnover rates are shown in Figures 3–5 and Table 3, respectively. The model parameters are listed in Table A2. The porewater geochemistry of the

hydrate-bearing hole W08B displayed that the SMTZ must be located near the seabed with depth shallower than 1 mbsf, but its exact depth cannot be resolved by the sampling scheme during the gas hydrate drilling expedition (Figure 3, Table 3). The $CH_4$ and $SO_4^{2-}$ profiles and methane turnover were instead constrained by the depth where gas hydrates first occurred and may represent the minimum values. Our simulations generally reproduced the measured concentrations of $SO_4^{2-}$, DIC and $Ca^{2+}$ in the investigated cores except the enigmatic reversals in R7-3 below 0.5 mbsf.

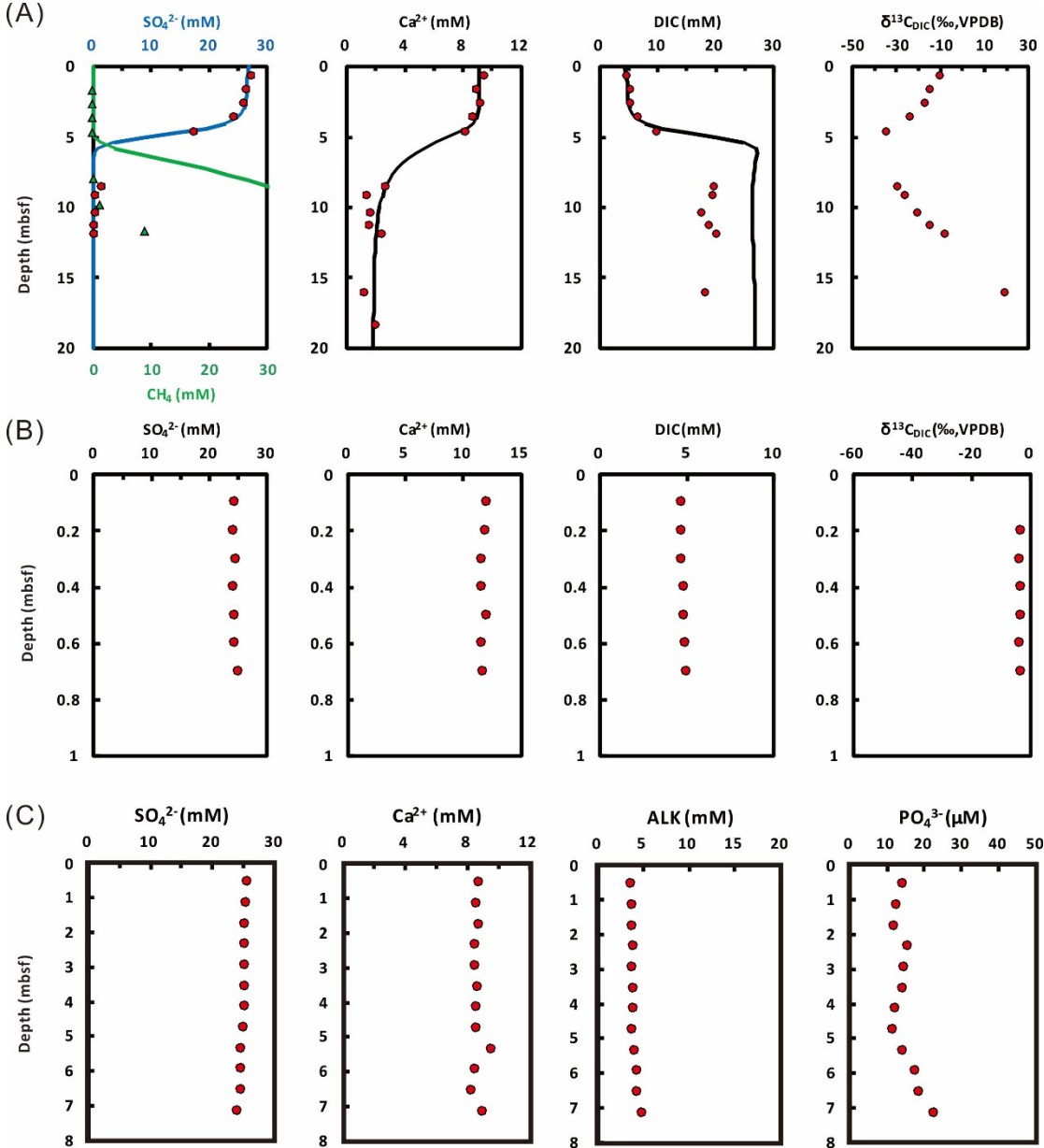

**Figure 4.** Measured (dots) and simulated (curves) depth profiles of cores in the seep area QH-ROV05. Down-depth concentration of $SO_4^{2-}$, $Ca^{2+}$, DIC, $\delta^{13}C_{DIC}$ and/or $CH_4$ of (**A**) W09 above 20 mbsf and (**B**) R5-1 are shown. Down-depth concentration of $SO_4^{2-}$, $Ca^{2+}$, total alkalinity (ALK) and $PO_4^{3-}$ of core CL48 are also shown (**C**).

The rates of POC degradation through sulfate reduction ranged between 0.1 and 1.8 mmol m$^{-2}$ yr$^{-1}$ at the study cores. Compared to the low depth-integrated rates of POC degradation, the AOM overwhelmingly dominated the depletion of sulfate with depth-integrated rates of 591, 383, 241, 410, 80.3 and 59.0 mmol m$^{-2}$ yr$^{-1}$ for cores W08B, W08C, W09, HM-1, R7-1 and R7-3, respectively. The

AOM rates were mainly sustained by an external methane source, and methanogenesis contributed only a negligible amount of methane (Table 3). Noting that the methane turnovers of W08B may be underestimated due to the lack of data in the surface sediments and the uncertainty of the exact depth of SMTZ. The benthic DIC fluxes were estimated to be 460, 272, 211, 295, 57.1 and 50.8 mmol m$^{-2}$ yr$^{-1}$ for cores W08B, W08C, W09, HM-1, R7-1 and R7-3, respectively (Table 3).

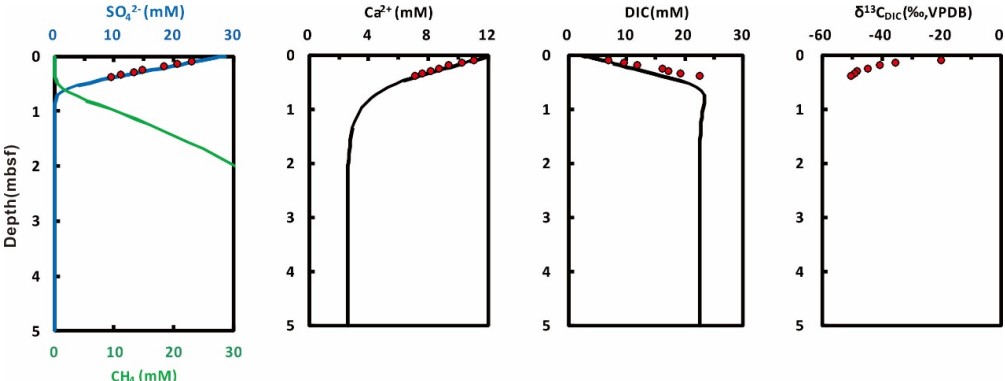

**Figure 5.** Measured (dots) and simulated (curves) depth profiles of core HM-1. Down-depth concentration of $SO_4^{2-}$, $CH_4$, $Ca^{2+}$, DIC and $\delta^{13}C_{DIC}$ are shown.

**Table 3.** Depth-integrated simulated turnover rates and benthic methane fluxes based on the numerical modelling.

| Depth-Integrated Flux | W08B | W08C | W09 | R7-1 | R7-3 | HM-1 | Unit |
|---|---|---|---|---|---|---|---|
| $F_{POC}$: Total POC mineralization rate | 2.0 | 2.6 | 5.9 | 1.1 | 1.1 | 1.5 | mmol m$^{-2}$ yr$^{-1}$ of C |
| $F_{OSR}$: Sulfate reduction via POC degradation | 0.1 | 0.8 | 1.8 | 0.3 | 0.4 | 0.3 | mmol m$^{-2}$ yr$^{-1}$ of $SO_4^{2-}$ |
| $F_{ME}$: Methanogenesis via POC degradation | 1.9 | 1.8 | 4 | 0.5 | 0.3 | 1.3 | mmol m$^{-2}$ yr$^{-1}$ of $CH_4$ |
| $F_{AOM}$: Anaerobic oxidation of methane | 591 | 383 | 241 | 80.3 | 59.0 | 378 | mmol m$^{-2}$ yr$^{-1}$ of $CH_4$ |
| $F_{CP}$: Authigenic $CaCO_3$ precipitation | 165 | 97.9 | 18.7 | 20.2 | 5.2 | 80.1 | mmol m$^{-2}$ yr$^{-1}$ of C |
| Dissolved $CH_4$ flux above GHOZ | 619 | 418 | 263 | 80.1 | 59.4 | 451 | mmol m$^{-2}$ yr$^{-1}$ of $CH_4$ |
| $CH_4$ efflux | 13.6 | 11.7 | 5.7 | $4.4\times10^{-4}$ | $4.4\times10^{-5}$ | 4.3 | mmol m$^{-2}$ yr$^{-1}$ of $CH_4$ |
| DIC efflux | 427 | 291 | 224 | 58.3 | 51.5 | 296 | mmol m$^{-2}$ yr$^{-1}$ of $CH_4$ |

## 5. Discussion

### 5.1. Methane-Related Carbon Cycling at Cold Seep Areas

In the Qiongdongnan Basin, upward expulsion of free gas in the sediment was identified by the blanking or pull-up seismic reflections in gas chimney and mud diapir structures [26,37,38]. Previous studies have shown that hydrocarbon seeps of W08 and W09 are characterized by abundant thermogenic gas within gas chimneys and a lack of advective fluid flow [28,39,40]. On the other hand, biogenic gas was transported upwards within fluid conduits and emitted to the water column at the stations HM-ROV and ROV2 in the eastern part of the Haima cold seeps [23,26]. The biogenic origin of emitted methane is also evident by extremely-depleted $^{13}C_{DIC}$ in porewater profiles of core HM-01 with the lowest $\delta^{13}C_{DIC}$ of $\sim-50‰$ (Figure 5; Table A4). In addition, the $\delta^{13}C_{DIC}$ values below the SMTZ became more positive (Figures 3–5), which is caused by the generation of $^{13}C$-enriched DIC via methanogenesis [52,53]. Nevertheless, modeling results display that in-situ methanogenesis rates in the upper sediments are too low to supply sufficient methane to form gas hydrate (Table 3). Hence, thermogenic or microbial gas transport from deep-seated sediments serves as the main methane source for AOM and hydrate formation in the shallow sediments of the investigated cores.

Our modeling results also show that, compared to sulfate reduction via POC degradation (OSR) throughout the sulfate reduction zone, AOM at the bottom of the sulfate reduction zone is the major pathway for the consumption of dissolved sulfate (Table 4). Noting that the methane-related carbon inventories are highly heterogeneous within short distances in an individual cold seep system. Outside of the seepage center within the gas chimney of QH-ROV07 (cores R7-1, R7-3), QH-ROV05 (CL48), HM-ROV (QDN-31) and HM-ROV05 (QDN-14A, QDN-14B), almost all the dissolved gas is consumed by AOM at lower rates, or there are no apparent upward flux of dissolved gas (Table 3; Table 4). In contrast, for the cores (W08B and W08C; W09; HM-01) within the seepage center, AOM rates are at least one order-of magnitude higher (Table 3; Table 4). Modeling results indicate that, above the GHOZ, AOM within the SMTZ is the main methane sink and also consume the majority of methane at these cores (Table 3). AOM increased porewater alkalinity by producing bicarbonate and led to authigenic carbonate precipitation as indicated by the decrease in $Ca^{2+}$ concentrations with depth (Figures 4–6).

**Table 4.** Comparison of methane turnover rates of various sites in Qiongdongnan Basin and other areas of the SCS.

| Site ID | $F_{SO4}$ | $F_{CH4}$ | $R_{AOM}$ | DIC Efflux | $Z_{SMTZ}$ (mbsf) | Profile above SMTZ | Reference |
|---|---|---|---|---|---|---|---|
| W08B | 592 | 619 | 591 | 427 | <0.5 | kink-type | |
| W08C | 393 | 418 | 383 | 291 | ~3 | kink-type | |
| W09 | 246 | 263 | 241 | 224 | ~7 | kink-type | This study |
| R7-1 | 80.6 | 80.1 | 80.3 | 58.3 | ~2.1 | linear | |
| R7-3 | 59.4 | 58.9 | 59.0 | 51.5 | ~3.1 | linear | |
| HM-1 | 378 | 389 | 410 | 327 | ~0.6 | linear | |
| R1 | 1226 | 4110 | 1225.7 | 1139 | 1.5 | kink-type | |
| QDN-14A | 450 | 540 | 449.3 | 404 | 3 | kink-type | [31] |
| QDN-14B | 193 | 507 | 193.1 | 131 | 5 | kink-type | |
| 2015XS-44 | 19.7 | 12.6 | 12.4 | 15 | 18.6 | kink-type | |
| 2015XS-50 | 31.9 | 25.8 | 24.6 | 25 | 18 | kink-type | [22] |
| 2015XS-R2 | 172 | 570.9 | 170.6 | 155 | 1.3 | kink-type | |
| CL30 | 39.3 | 31.4 | 35.3 | 21.7 | 4.7 | kink-type | |
| CL44 | 98.3 | 73.3 | 74.3 | 87 | 7 | kink-type | [30] |
| CL47 | 110 | 84.8 | 85 | 115 | 6.8 | kink-type | |
| 1PC | 59.5 | 59.5 | | | 7 | linear | [54] |
| C14 | 56 | 15.7 | 11 | 55 | 14.3 | linear | [46] |
| Shenhu | 2.0–40.0 | 2.0–37.0 | 7.8–30.5 | 10.1–31.7 | 7.7–87.9 | linear | [22,55] |
| Dongsha | 5.7–102 | 1.0–101.5 | 1.0–101.5 | 13.1–26.1 | 0.05–21.8 | linear and kink-type | [22,56] |
| Beikang | 34.5–62.7 | 24.5–62.7 | 27.5–43.1 | 32.3–50.1 | 5.3–8.8 | linear and kink-type | [57] |

Notation: $F_{SO4}$ and $F_{CH4}$ are the downward flux of $SO_4^{2-}$ and the upward flux of $CH_4$, while $R_{AOM}$ refers to the depth-integrated reaction rate of AOM (unit: mmol m$^{-2}$ yr$^{-1}$). $Z_{SMTZ}$ is the depth of SMTZ.

Comparing our model results with those of other sites in the Qiongdongnan Basin and in other areas of the SCS, the methane turnover rates in shallow sediments are much higher at the cold seeps in the Qiongdongnan Basin than those at a dormant pockmark in the nearby SW Xisha Uplift, as well as those at sites located on deep-seated gas hydrate reservoirs in the Shenhu area and the Beikang Basin (Table 4). Besides, the methane turnover rates at the active cold seep sites in the Taixinan Basin are of the same order of magnitude as those in the Qiongdongnan Basin (Table 4). The porewater profiles in the shallow sediments in these two basins often showed kink-type shape, which are attributed to irrigation of gas bubbling or recent increase in upward methane flux related to non-steady states of fluid seepage [20,30,45]. Considering the geological settings of the sampling sites, it is suggested that differences in the proximity of the sites to the fluid conduits including fractures, faults or anticlines mainly account for the variability in upward methane fluxes and turnover rates [20,28,31,39,45]. Overall, combing seafloor observations with geochemical modeling, our results further demonstrate that the methane-related carbon inventories are highly heterogeneous at different cold seep systems or different parts within short distances in an individual cold seep system. This signature should be attributed to the difference in the distance between the sampling sites and the seepage center.

### 5.2. Potential Contribution of Fossil Carbon from Cold Seeps to Bottom-Water Carbon Pool

Cold seep systems contribute considerable proportions of the local carbon budget of the overlying water column by emitting large amount of fossil carbon with depleted $^{13}$C and $^{14}$C into the water column. The seepage often results in a significant decrease in the $\delta^{13}C_{DIC}$ and $\Delta^{14}C_{DIC}$, and a small increase in the DIC in the overlying seawater, either by in-situ oxidation of vent methane or the concurrent input of DIC from seeps, or both [10,58–61]. Generally, the study of the DIC system in seawater can be greatly simplified by describing the system in terms of the alkalinity. In shallow sediments where pH is between 7.1 and 8.1, total alkalinity is often treated as carbonate alkalinity by ignoring other minor species for practical purposes [62].

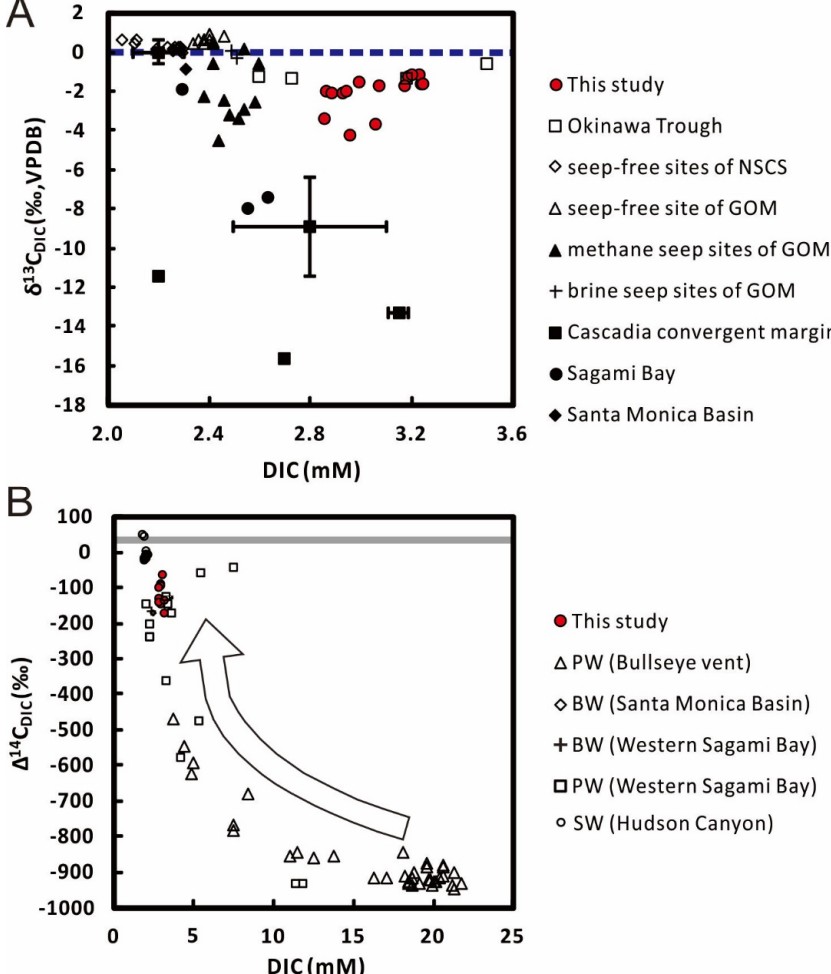

**Figure 6.** (**A**) DIC concentration versus $\delta^{13}C_{DIC}$ values of the bottom water (and overlying water column) from cold seep sites and seep-free sites worldwide. (**B**) DIC concentration versus $\Delta^{14}C_{DIC}$ of the bottom water and porewater from cold seep sites and seep-free sites globally. The blue dash line in A represents the background $\delta^{13}C_{DIC}$ value of seawater of South China Sea (SCS) [63], whereas the shade in B represents the background $\Delta^{14}C_{DIC}$ values of seawater of SCS [64]. The small increase in DIC in the seep sites of the Gulf of Mexico (GOM) and Cascadia convergent margin was attributed to the removal of DIC by active deposition of authigenic carbonates [10,60]. The dataset includes those of the bottom water (and overlying water column) and/or porewater at the seep sites of the Okinawa Trough [65], GOM [10,66], Hudson Canyon [61], Cascadia convergent margin [60], Western Sagami Bay [58,59], Santa Monica Basin [67] and Bullseye vent [68], as well as the seep-free sites of the NSCS [63].

Studies show that the benthic DIC fluxes at the SWI are in the order of magnitude of 10–10$^3$ mmol m$^{-2}$ yr$^{-1}$ in the cold seep areas of the Qiongdongnan Basin (Table 4). In addition, compared with

other seep sites, the slightly elevated alkalinity concentrations together with negative $\delta^{13}C$ values of bottom water suggest cold seep system may have influenced the carbon pool of bottom water (Figure 6, Table 4). In addition, the depletion in $^{14}C$ of DIC (−178‰ to −71‰) suggests that oxidation of $CH_4$ from deeper reservoirs is likely the source of ancient carbon to bottom waters (Figure 6B, Table A5).

To understand the impact of a seep-derived source of TA (excess alkalinity) on deep water inorganic carbon pool, the following simple two-endmember mass balance model is used:

$$TA_{wc}\delta^{13}C_{wc} = TA_{ex}\delta^{13}C_{ex} + TA_{bg}\delta^{13}C_{bg}, \tag{4}$$

$$TA_{wc} = TA_{ex} + TA_{bg}, \tag{5}$$

where the subscript "wc", "ex" and "bg" stand for the water column, excess TA, and background value of the SCS, respectively. The background concentration and $\delta^{13}C$ of TA in the bottom water of northern SCS are ca. 2.4 mM and 0 ‰, respectively [63]. By inserting the background and our measured values of TA into the above two equations, we estimate the $TA_{ex}$ and its carbon isotope composition are approximately 0.5–0.8 mM and −22.7‰ to −4.0‰, respectively (Figure 6; Table 4). The $\delta^{13}C$ compositions of the excess carbonate alkalinity are similar to those of the pore fluids at the top of the push core near the cold seep vent (−19.8‰ at HM-1, Table A3). The excess $\delta^{13}C$ values of DIC are to be comparable to those of vent fluids or uppermost pore fluids on the Cascadia convergent margin (−15.6‰ to −3.9‰) [60], on the slope of Gulf of Mexico (−27.8‰) [10], and in the Guaymas Basin (−25.6‰) [69]. These characteristics support the hypothesis that the carbonate alkalinity and $\delta^{13}C_{DIC}$ anomalies in the water samples result from the mixture of seawater and cold seep fluids emitting on the seafloor.

Moreover, the bottom-water samples with the lowest $\delta^{13}C_{DIC}$ values were collected in the stations with living or recently dead chemosynthetic bivalves, further indicating that high-flux seep sites serve as an important source of DIC to the water column (Figure 6A, Table A5).Model results show that benthic DIC fluxes in the study area are $10^1$-$10^3$ mmol m$^{-2}$ a$^{-1}$, which is in the range of those reported at other cold seep sites in the SCS [22,30,45,46,56,57]. We postulate that the DIC released from seep-impacted sediments could alter the dissolved carbon pool in the overlying bottom water. To further quantify the local DIC contributions from cold seeps in the study area, we estimate the ratio of the excess alkalinity to the alkalinity of the bottom-water samples. The calculations yield the contribution of DIC from cold seep fluids to the total bottom water ranging between 16% and 26% (Table A5). This is similar to the estimates in the Gulf of Mexico and Okinawa Trough where cold seep fluids can contribute up to 14.3% and 20% of excess DIC to the bottom water, respectively [10,65]. Hydroacoustic imaging revealed that the height of the gas plume was ~770 m at seep C within the Haima cold seeps [26], yet the contribution of cold seep fluids to the water column is still unknown. Here our calculation suggests that the upward DIC flux from the cold seeps could contribute up to one quarter of $^{13}C$-depleted DIC to the bottom-water inorganic carbon pool. Therefore, the impact of cold seep fluids to the overlying water column inorganic carbon reservoir seems to be significant.

Previous investigation has showed that the area of seepage at the Haima cold seeps is approximately 350 km$^2$ [70]. Based on the cores sampled from this area by far, the model-derived benthic DIC fluxes in this area ranged between 15 and 1,139 mmol m$^{-2}$ yr$^{-1}$ with an average of 239 mmol m$^{-2}$ yr$^{-1}$ (Table 4). By multiplying this value by the area of seepage, the average DIC efflux amounts to ~$8.4 \times 10^{-5}$ Tmol yr$^{-1}$ at this active cold seeps. The amount of DIC derived from the cold seeps is likely greater than our estimation because the DIC efflux is highly heterogeneous in a cold seep area. It is suggested that release of DIC into bottom water can, in some cases, promote the production and preservation of biogenic and authigenic carbonate. Nevertheless, aerobic oxidation of methane from subseafloor can produce $CO_2$ and lower the seawater pH, thereby probably dissolving carbonate [71]. A previous study showed that, at least ~$7 \times 10^{-4}$ Tmol DIC was released from marine sediments per year in cold seeps and hydrate-bearing areas assuming an area of $1.6 \times 10^4$ km$^2$ in the northern SCS [22]. Compared to this estimation, our results suggest that DIC from the active Haima cold seeps probably represent an important source of fossil carbon to the overlying water column. However, more work is required

to understand the influence of methane seepage on the chemistry of bottom-water carbonate system, especially the amount of DIC released to the hydrosphere.

## 6. Conclusions

The geochemical composition of porewater in shallow sediments and bottom seawater were investigated at four cold seeps in the northwestern SCS. As a result of the study, we conclude the following:

1. Thermogenic or microbial gas transport from deep-seated sediments serves as the main methane source and dissolved at high-flux sites near the seepage centers. Most dissolved methane is consumed by AOM in the cores W08B, W08C, W09, R7-1, R7-3 and HM-1 as indicated by the shallow SMTZ (~7 mbsf to <0.5 mbsf) and our model results. Depth-integrated AOM rates range from 59.0 to 591 mmol $m^{-2}$ $yr^{-1}$. The methane-related carbon turnovers are highly heterogeneous at the studied cold seep systems. We attribute this heterogeneity at cold seeps to the difference in proximity of the sampling sites to the center of fluid conduit.

2. The DIC effluxes range from 51.5 to 427 mmol $m^{-2}$ $yr^{-1}$ at the study sites. [13]C- and [14]C-depleted fossil DIC in seep fluids may be released to the bottom water at the four seep areas based on the lower $\delta^{13}C$ and $\Delta^{14}C$ values of bottom-water DIC. Simplified estimations show that cold seeps fluids may contribute 16–26% of the bottom seawater DIC. In addition, a rough average $8.4 \times 10^{-5}$ Tmol DIC may be released from shallow sediments to the water column annually at the Haima cold seeps.

Overall, this study shows that the contribution of cold seep fluids is significant both to the pore fluids and the bottom seawater, and may have considerable impacts on the carbon cycle as well as on the seafloor chemosynthetic ecosystems in the cold seep areas.

**Author Contributions:** Conceptualization—J.F. and N.L.; methodology—J.F. and N.L.; formal analysis—J.F., N.L. and M.L.; resources—J.L., S.Y., H.W., D.C.; writing—original draft preparation —J.F. and N.L.; writing—review and editing —J.F. and M.L.; funding acquisition —S.Y., H.W., J.L. All authors have read and agreed to the published version of the manuscript.

**Funding:** This research was funded by the National Key R&D Program of China (Grant: 2018YFC0310004, 2018YFC0310006), the open-funds of Key Laboratory of Marine Mineral Resources, Ministry of Land and Resources (Grant: KLMMR-2017-A-08), the National Natural Science Foundation of China (Grant: 41730528, 41806074, 41706053, 41976061), the National Special Project on Gas Hydrate of China (Grant: GZH201100301), the Major Program of Guangdong Basic and Applied Research (Grant: 2019B030302004), the Major Program of Guangdong Laboratory for Marine Science and Engineering of South China (Guangzhou) (Grant: GML2019ZD01) and Guangzhou Science and Technology (Grant: 201909010002).

**Acknowledgments:** We thank the crew of the Haiyang-6 exploration ship for collecting the sediment cores. We are also grateful to our colleagues from GMGS and Zhejiang University for their collection and analysis to the porewater and bottom-water samples. We also thank all the participants in the GMGS5 expedition for their support. The authors appreciate Yongbo Peng (Louisiana State University) for the help with the geochemical analyses.

**Conflicts of Interest:** The authors declare that they have no conflicts of interest. The funding sponsors had no role in the design of the study; in the collection, analyses, or interpretation of data; in the writing of the manuscript; or in the decision to publish the results.

## Appendix A

The model solved the following partial differential equations for solid and dissolved species, respectively [72,73]:

$$(1 - \Phi)\frac{\partial C_s}{\partial t} = -\frac{\partial((1 - \Phi)\cdot v_s \cdot C_s)}{\partial x} + (1 - \Phi)\cdot \Sigma R, \tag{A1}$$

$$\Phi\frac{\partial C_a}{\partial t} = \frac{\partial\left(\Phi\cdot D_s\cdot\frac{\partial C_a}{\partial x}\right)}{\partial x} - \frac{\partial\left(\Phi\cdot v_p\cdot C_a\right)}{\partial x} + \Phi\cdot\Sigma R, \tag{A2}$$

where $x$ (cm) is depth, $t$ (yr) is time, $\Phi$ is porosity, $D_s$ ($cm^2$ $yr^{-1}$) is the molecular diffusion coefficient corrected for tortuosity, $C_a$ ($\mu mol$ $cm^{-3}$) is the concentration of dissolved species, POC is in wt.%, $v_p$ (cm $yr^{-1}$) is the burial velocity of porewater, $v_s$ (cm $yr^{-1}$) is the burial velocity of solids and $\Sigma R$ denotes the sum of the rates of biogeochemical reactions considered in the model.

Since the porosity of our studies cores is not available, we took the mean porosity from an adjacent core [74] and applied it in the model run. As a result, sediment compaction was neglected, and $v_s$ and $v_p$ were equivalent to the sedimentation rate ($\omega$) which is approximated according to [74,75].

Depth-dependent molecular diffusion coefficients of dissolved species were calculated after [73] and [76] and $D_s$ were corrected for tortuosity:

$$D_s = \frac{D_m}{1 - ln(\Phi)^2},\tag{A3}$$

where $D_m$ is the molecular diffusion coefficient in free seawater at the in-situ temperature, salinity and pressure values. The diffusive transport of DIC was simulated using the diffusion coefficient of bicarbonate ($HCO_3^-$) since this is the dominant anion.

The major reactions considered in the model are particulate organic matter (POM) degradation via sulfate reduction, methanogenesis, AOM and authigenic carbonate precipitation. The net reaction terms of the one solid (POC) and four dissolved species ($SO_4^{2-}$, $CH_4$, DIC and $Ca^{2+}$) are shown in Table A1. All the model parameters are given in Table A2.

The length of the simulated cores was set to 1000 cm. Upper boundary conditions for all species were imposed as fixed concentrations (Dirichlet boundary) using measured values in the uppermost sediment layer. A zero concentration gradient (Neumann-type boundary) was imposed at the lower boundary for all species. The model was solved using the NDSolve object of MATHEMATICA V. 10.0. All simulations were run for $10^7$ years to achieve the steady state with a mass conservation of >99 %.

**Table A1.** Reaction terms of all species used in the model.

| Species | Rate |
|---|---|
| Particulate organic carbon (POC) | $-R_{POC}$ |
| Sulfate ($SO_4^{2-}$) | $-R_{SR} - R_{AOM}$ |
| Methane ($CH_4$) | $-R_{AOM} + R_{MG}$ |
| Dissolved inorganic carbon (DIC) | $R_{POC}/f_{POC} - R_{MG} + R_{AOM} - R_{CP}$ |
| Calcium ($Ca^{2+}$) | $-R_{CP}$ |

**Table A2.** Summary of model parameters and boundary conditions used in the model simulations.

| Parameter | W08B | W08C | W09 | R7-1 | R7-3 | HM-1 | Unit |
|---|---|---|---|---|---|---|---|
| Temperature ($T$)[a] | 2.3 | 2.3 | 2.3 | 2.3 | 2.3 | 2.9 | °C |
| Salinity ($S$) | 34 | 34 | 34 | 34 | 34 | 34 | PSU |
| Pressure ($P$)[b] | 17.7 | 17.7 | 17.6 | 17.7 | 17.7 | 14.2 | MPa |
| Density of dry solids ($\rho_s$)[c] | 2.6 | 2.6 | 2.6 | 2.6 | 2.6 | 2.6 | g cm$^{-3}$ |
| Density of porewater ($\rho_{pw}$)[c] | 1.033 | 1.033 | 1.033 | 1.033 | 1.033 | 1.033 | g cm$^{-3}$ |
| Sedimentation rate ($\omega$)[d] | 0.017 | 0.017 | 0.017 | 0.017 | 0.017 | 0.017 | cm yr$^{-1}$ |
| Porosity ($\Phi$)[e] | 0.7 | 0.7 | 0.7 | 0.7 | 0.7 | 0.7 | - |
| Initial age of POC ($a_0$)[f] | 950 | 950 | 950 | 950 | 950 | 650 | kyr |
| Molecular diffusion coefficient of $SO_4^{2-}$ in free seawater[g] | 191 | 191 | 191 | 191 | 191 | 191 | cm$^2$ yr$^{-1}$ |
| Molecular diffusion coefficient of $CH_4$ in free seawater[g] | 294 | 294 | 294 | 294 | 294 | 294 | cm$^2$ yr$^{-1}$ |
| Molecular diffusion coefficient of DIC in free seawater[g] | 203 | 203 | 203 | 203 | 203 | 203 | cm$^2$ yr$^{-1}$ |
| Molecular diffusion coefficient of $Ca^{2+}$ in free seawater[g] | 142 | 142 | 142 | 142 | 142 | 142 | cm$^2$ yr$^{-1}$ |
| Michaelis–Menten constant for POC degradation ($K_{SO_4^{2-}}$)[h] | $1 \times 10^{-4}$ | $1 \times 10^{-4}$ | $1 \times 10^{-4}$ | $1 \times 10^{-4}$ | $1 \times 10^{-4}$ | $1 \times 10^{-4}$ | mM |
| Rate constant for AOM ($k_{AOM}$) | 400 | 50 | 50 | 50 | 50 | 400 | cm$^3$ yr$^{-1}$ mmol$^{-1}$ |
| Rate constant for carbonate precipitation/dissolution ($k_{Ca}$) | $1.2 \times 10^{-3}$ | $6.0 \times 10^{-6}$ | $2.0 \times 10^{-7}$ | $1.0 \times 10^{-6}$ | $1.0 \times 10^{-7}$ | $1.1 \times 10^{-5}$ | mmol cm$^{-3}$ yr$^{-1}$ |
| Upper boundary condition for POC | 1.2 | 1.2 | 1.3 | 1.2 | 1.2 | 0.9 | wt.% |
| Upper boundary condition for $SO_4^{2-}$ | 25 | 25 | 27 | 24 | 25 | 29 | mM |
| Upper boundary condition for DIC | 4.6 | 4.6 | 4.8 | 5.3 | 4.8 | 3.5 | mM |
| Upper boundary condition for $Ca^{2+}$ | 12.5 | 12.1 | 9.1 | 12.0 | 11.8 | 12.5 | μM |
| Upper boundary condition for $CH_4$ | 0 | 0 | 0 | 0 | 0 | 0 | mM |
| Lower boundary condition for $SO_4^{2-}$ | $\partial C/\partial x = 0$ | $\partial C/\partial x = 0$ | $\partial C/\partial x = 0$ | $\partial C/\partial x = 0$ | $\partial C/\partial x = 0$ | $\partial C/\partial x = 0$ | - |
| Lower boundary condition for DIC | $\partial C/\partial x = 0$ | $\partial C/\partial x = 0$ | $\partial C/\partial x = 0$ | $\partial C/\partial x = 0$ | $\partial C/\partial x = 0$ | $\partial C/\partial x = 0$ | - |
| Lower boundary condition for $Ca^{2+}$ | $\partial C/\partial x = 0$ | $\partial C/\partial x = 0$ | $\partial C/\partial x = 0$ | $\partial C/\partial x = 0$ | $\partial C/\partial x = 0$ | $\partial C/\partial x = 0$ | - |
| Lower boundary condition for $CH_4$ | - | - | - | 13 | 4.5 | - | mM |

[a] Estimated according to the temperature data obtained during ROV cruise, [b] calculated from water depth, [c] [30], [d] [74,75], [e] modified after [74], [f] adopted from parameters at cores CL48 and 2015XS-R2 [22], [g] according to [77], [h] [44].

## Appendix B

Appendix B includes full data of analytical results of the porewater samples of the studied cores (Tables A3 and A4) and the bottom seawater samples (Table A5).

**Table A3.** Concentrations and isotope ratios of various ions from the upper sediments of the boreholes W08B, W08C and W09.

| Depth (cmbsf) | $SO_4^{2-}$ (mM) | $Ca^{2+}$ (mM) | DIC (mM) | $\delta^{13}C_{DIC}$ (‰,VPDB) | Depth (cmbsf) | $CH_4$ (mM) |
|---|---|---|---|---|---|---|
| W08B | | | | | | |
| 47 | 0.7 | 2.7 | 20.3 | −13.5 | - | - |
| 114 | 1.2 | 2.7 | 21.2 | −13.4 | 102 | 9 |
| 196 | 0.4 | 3.0 | 19.4 | −7.6 | - | - |
| 255 | 0.6 | 2.2 | 17.2 | −9.8 | 253 | 3.6 |
| 361 | 0.2 | - | - | - | - | - |
| 441 | 0.4 | - | - | - | - | - |
| 835 | 0.3 | 2.7 | 14.9 | 9.8 | 835 | 15.3 |
| 935 | 0.1 | 2.7 | 18.9 | 11.6 | 925 | 4.4 |
| 990 | 0.1 | 2.7 | 18.7 | 12.1 | - | - |
| W08C | | | | | | |
| 170 | 22.5 | 2.2 | -21.1 | 8.6 | 50 | 2.8 |
| 796 | 0.1 | - | - | - | - | - |
| 880 | 1.6 | 2.6 | 12.5 | 18.5 | 890 | 6.2 |
| 963 | 0.7 | - | - | - | - | - |
| W09 | | | | | | |
| 60 | 27.3 | 9.5 | 4.7 | −10.0 | - | - |
| 160 | 26.4 | 9.0 | 5.3 | −14.4 | 170 | 0 |
| 260 | 25.9 | 9.2 | 5.4 | −16.9 | 270 | 0 |
| 360 | 24.2 | 8.7 | 6.7 | −23.9 | 370 | 0 |
| 460 | 17.3 | 8.2 | 9.9 | −34.4 | 470 | 0 |
| 851 | 1.5 | 2.7 | 19.7 | −29.6 | 800 | 0.2 |
| 919 | 0.4 | 1.4 | 19.6 | −26.2 | 984 | 1.2 |
| 1036 | 0.4 | 1.7 | 17.6 | −20.4 | - | - |
| 1127 | 0.2 | 1.6 | 19.0 | −14.9 | - | - |
| 1189 | 0.2 | 2.5 | 20.3 | −8.0 | 1176 | 8.9 |
| 1610 | 3.9 | 1.2 | 18.3 | 19.3 | - | - |
| 1836 | 2.6 | 2.0 | - | - | - | - |

**Table A4.** Concentrations and isotope ratios of various dissolved components at the studied push cores and piston core.

| Depth (cmbsf) | $SO_4^{2-}$ (mM) | $Ca^{2+}$ (mM) | DIC (mM) | $\delta^{13}C_{DIC}$ (‰,VPDB) |
|---|---|---|---|---|
| R5-1 | | | | |
| 10 | 24.4 | 12.1 | 4.7 | - |
| 20 | 24.3 | 12.0 | 4.7 | −3.1 |
| 30 | 24.6 | 11.7 | 4.7 | −3.6 |
| 40 | 24.3 | 11.6 | 4.8 | −3.0 |
| 50 | 24.5 | 12.1 | 4.9 | −3.3 |
| 60 | 24.4 | 11.6 | 4.9 | −3.6 |
| 70 | 25.1 | 11.7 | 5.0 | −3.2 |
| R7 | | | | |
| 10 | 24.0 | 12.1 | - | - |
| 20 | 23.5 | 11.7 | 4.7 | −1.8 |
| 30 | 23.3 | 11.6 | 4.3 | −3.0 |
| 40 | 23.7 | 11.7 | 5.1 | −9.3 |
| 50 | 23.5 | 12.3 | 4.4 | −4.9 |
| 60 | 22.9 | 11.3 | 4.3 | −4.6 |

**Table A4.** *Cont.*

| Depth (cmbsf) | $SO_4^{2-}$ (mM) | $Ca^{2+}$ (mM) | DIC (mM) | $\delta^{13}C_{DIC}$ (‰,VPDB) |
|---|---|---|---|---|
| R7-1 | | | | |
| 5 | 23.5 | 11.6 | 6.0 | −7.5 |
| 10 | 22.7 | 11.6 | 6.4 | −10.4 |
| 15 | 22.1 | 11.5 | 6.8 | −13.1 |
| 20 | 22.6 | 13.4 | 6.9 | −14.0 |
| 25 | 21.2 | 10.9 | 5.3 | −16.2 |
| 30 | 19.8 | 10.7 | 4.6 | −18.5 |
| 35 | 19.4 | 10.7 | 11.2 | −29.8 |
| 40 | 20.8 | 11.2 | 10.3 | −30.5 |
| 45 | 19.4 | 11.0 | 11.1 | −32.7 |
| 50 | 19.1 | 10.1 | 11.1 | −32.5 |
| 55 | 18.0 | 8.9 | 12.0 | −33.3 |
| R7-2 | | | | |
| 5 | 24.7 | 12.1 | 4.7 | −2.3 |
| 10 | 24.3 | 11.8 | 5.4 | −2.9 |
| 15 | 24.2 | 11.8 | 4.7 | −2.9 |
| 20 | 24.6 | 12.3 | 4.7 | −2.3 |
| 25 | 24.7 | 12.0 | 4.6 | −2.2 |
| 30 | - | - | 4.7 | −7.0 |
| 35 | 24.1 | 11.8 | 4.2 | −3.3 |
| 40 | 23.8 | 12.0 | 5.3 | −4.1 |
| 45 | 24.4 | 11.6 | 4.7 | −5.1 |
| 50 | 24.5 | 11.8 | 0.5 | −7.8 |
| 55 | 24.3 | 11.8 | - | - |
| R7-3 | | | | |
| 5 | 24.5 | 12.5 | 4.8 | −2.3 |
| 10 | 24.2 | 11.8 | - | - |
| 15 | 23.8 | 11.7 | 6.6 | −8.8 |
| 20 | 23.0 | 11.8 | - | - |
| 25 | 22.9 | 11.7 | 7.2 | −12.8 |
| 30 | 22.4 | 11.7 | - | - |
| 35 | 22.4 | 11.3 | 7.0 | −17.2 |
| 40 | 21.7 | 11.6 | - | - |
| 45 | 21.8 | 11.4 | 8.4 | −22.7 |
| 50 | 22.0 | 11.2 | - | - |
| 55 | 22.2 | 11.3 | 7.3 | −23.3 |
| 60 | 22.5 | 11.7 | - | - |
| 65 | 22.5 | 11.6 | 7.3 | −22.5 |
| 70 | 23.0 | 11.5 | - | - |
| HM-1 | | | | |
| 10 | 23.1 | 11.1 | 7.0 | −19.8 |
| 15 | 20.7 | 10.3 | 9.7 | −35.2 |
| 20 | 18.5 | 9.4 | 11.9 | −40.6 |
| 25 | 14.9 | 8.7 | 16.2 | −44.3 |
| 30 | 13.4 | 8.2 | 17.4 | −48.2 |
| 35 | 11.3 | 7.6 | 19.4 | −48.8 |
| 40 | 9.5 | 7.1 | 22.7 | −49.9 |

| Depth (cmbsf) | $SO_4^{2-}$ (mM) | $Ca^{2+}$ (mM) | Alk (mM) | $PO_4^{3-}$ (μM) |
|---|---|---|---|---|
| CL48 | | | | |
| 55 | 25.7 | 8.7 | 3.6 | 14.3 |
| 115 | 25.5 | 8.6 | 3.8 | 12.8 |
| 175 | 25.3 | 8.8 | 3.8 | 12.0 |
| 235 | 25.4 | 8.5 | 3.9 | 15.5 |
| 295 | 25.3 | 8.5 | 3.9 | 14.5 |

**Table A4.** *Cont.*

| Depth (cmbsf) | $SO_4^{2-}$ (mM) | $Ca^{2+}$ (mM) | Alk (mM) | $PO_4^{3-}$ (μM) |
|---|---|---|---|---|
| 355 | 25.3 | 8.7 | 3.9 | 14.3 |
| 415 | 25.3 | 8.6 | 3.9 | 12.3 |
| 475 | 25.2 | 8.6 | 3.9 | 11.5 |
| 535 | 24.7 | 9.6 | 4.0 | 14.3 |
| 595 | 24.6 | 8.5 | 4.3 | 17.6 |
| 655 | 24.7 | 8.3 | 4.4 | 18.8 |
| 715 | 24.1 | 9.0 | 4.9 | 22.6 |

**Table A5.** Geochemical data of bottom seawater and the isotope mass-balance model results.

| Site | Samle ID | Sampling Method | pH | TA (mM) | $\delta^{13}C_{DIC}$ (‰,VPDB) | $^{14}C$ age (yr BP) | $\Delta^{14}C$ (‰,VPDB) | $\delta^{13}C_{CS}$ (‰,VPDB) | Fcs (%) | $Alk_{MD}$ (mM) |
|---|---|---|---|---|---|---|---|---|---|---|
| ROV05 | R-05-shell | T,P-tight | 7.7 | 3.1 | −3.7 | 800 | −95 | −17.2 | 21.6 | 0.7 |
| ROV05 | ROV05 | CTD | 7.7 | 3.2 | −1.6 | 1230 | −142 | −6.3 | 25.9 | 0.8 |
| ROV05 | ROV05-1 | Water on the top of the tubes | 7.7 | 3.2 | −1.1 | | | −4.6 | 25.0 | 0.8 |
| R7 | R-07 | T,P-tight | 7.6 | 2.9 | −2.0 | 1170 | −136 | −12.5 | 16.1 | 0.5 |
| R7 | ROV07 | CTD | 7.7 | 3.2 | −1.1 | 1580 | −178 | −4.4 | 25.8 | 0.8 |
| R7 | ROV07+v | Water on the top of the tubes | 7.8 | 2.9 | −2.0 | | | −11.1 | 18.4 | 0.5 |
| ROV7-1 | R-07-1 | T,P-tight | 7.9 | 3.0 | −4.3 | 860 | −101 | −22.7 | 18.8 | 0.6 |
| ROV7-1 | R01-2018 | CTD | 7.9 | 3.1 | −1.8 | 590 | −71 | −8.0 | 22.0 | 0.7 |
| ROV7-1 | ROV07-1 | Water on the top of the tubes | 7.8 | 3.2 | −1.7 | | | −7.1 | 24.4 | 0.8 |
| HM-ROV | HM-2-vent | T,P-tight | 7.7 | 2.9 | −2.1 | 880 | −104 | −12.3 | 16.8 | 0.5 |
| HM-ROV | HM-3-vent | T,P-tight | 7.7 | 2.9 | −3.4 | 1250 | −144 | −21.2 | 16.1 | 0.5 |
| HM-ROV | HM-2 | CTD | 7.9 | 2.9 | −2.1 | | | −11.5 | 18.0 | 0.5 |
| HM-ROV | HM-3 | CTD | 7.7 | 3.2 | −1.3 | | | −5.2 | 24.8 | 0.8 |
| HM-ROV05-1 | HM-R003-1 | T,P-tight | 7.7 | 3.0 | −1.5 | 1300 | −149 | −7.5 | 19.9 | 0.6 |
| HM-ROV05-1 | HM-1 | CTD | 7.7 | 3.2 | −1.7 | | | −6.4 | 26.0 | 0.8 |

Notation: $\delta^{13}C_{CS}$ is the $\delta^{13}C_{TA}$ values of the cold seep endmember. fcs is the fractional contributions of the cold seep endmember to the deep-seawater alkalinity. $Alk_{MD}$ refers to the difference between the $\delta^{13}C_{TA}$ values of the bottom-water sample and the background value. CTD refers to the conductivity–temperature–depth instrument.

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
