# Peer review of "A Quantitative Assessment of Methane-Derived Carbon Cycling at the Cold Seeps in the Northwestern South China Sea"

_minerals, doi:10.3390/min10030256_

Round 1
Reviewer 1 Report
Dear Editor,
I read this manuscript with great interest. The work addresses a very important problem related to the influence of carbon from methane seeps on the overall carbon cycle. The manuscript has been submitted to the journal Minerals, however, it does not actually say anything about minerals, in particular authigenic carbonates, which "take" carbon from methane discharged from seeps. However, much attention is paid to the important product of the oxidation of methane and organic matter from which carbonates crystallize - dissolved inorganic carbon. I believe that this work can be published in the Minerals after minor changes and clarifications.
Rows 162-165 - You measured the gas that was obtained from the sediments by heating. In this case you likely get interstitial gas + gas bound to mineral or/and organic surfaces. This gas will differ in isotopic and molecular composition from the gas obtained by the “head space” method, which is most common in the study of marine bottom sediments. Please explain in more detail why you chose to heat the sediments rather than use the more common headspace method.
Row 188 (Table 2): Please decipher the meaning of all components used in the equations.
Row 196: Why did you use diffusion rather than fluid filtration for seepage environments?
Rows 217-220: In “Redfield” equation for organic matter the ratio of C:N:P = 106:16:1. Are you really sure that phosphorus should be used to balance the amount of CO2 and CH4?
Rows 254-262 (Figure 2): B – Please mark the massive authigenic carbonate by arrow, if possible.
Row 318: reversal in R7-3 seems only for sulfate, not for DIC and Ca
Row 353: Modeling results indicate that…
Row 374 (Table 4): Decipher the meaning of F and R and zsmt (may be smtz?)
Rows 419-421: You describe the delta 13C DIC, and refer to Figure 6B, where the delta 14C values are given.
Row 463: Perhaps, DIC effluxes must be up to 492 mmol m-2yr-1, not 427 (based on table 4)
Row 511: Perhaps, tables A1 and A2, not the S1 and S2? I didn’t find the tables marked “S”.
Reviewer 2 Report
The manuscript is a comprehensive study of methane turnover in cold seeps. Here are some points which should be cleared.
It is better to be mentioned some studies had been done in methane seepages.
In line 155, the authors mentioned a method with UV-vis and referred a reference but in the mentioned reference, there is not that method.
In table 2 the kinetic rate law for sulfate reduction is the same as methanogenesis which is wrong.
The measured and simulated profiles of POC for each core should be reported.
Round 2
Reviewer 1 Report
I am happy with revision made by authors.
Reviewer 2 Report
The manuscript is a comprehensive study of methane turnover in cold seeps